Identification of key genes and pathways associated with cholangiocarcinoma development based on weighted gene correlation network analysis

Liu Jingwei
Liu Weixin chunchang88@sohu.com
Li Hao
Deng Qiuping
Yang Meiqi
Li Xuemei
Liang Zeng
Department of Gastroenterology, The First Affiliated Hospital of China Medical University , Shenyang, Liaoning , China
Ghouri Yezaz
Electronic publication date: 2019 Oct 31
Publication date: 2019
Volume: 7
Electronic Location ID: e7968
Received 2019 May 28; Accepted 2019 Oct 1
Copyright: © 2019 Liu et al.
Copyright year: 2019
Copyright holder: Liu et al.
License: This is an open access article distributed under the terms of the Creative Commons Attribution License, which permits unrestricted use, distribution, reproduction and adaptation in any medium and for any purpose provided that it is properly attributed. For attribution, the original author(s), title, publication source (PeerJ) and either DOI or URL of the article must be cited.
License URL: https://creativecommons.org/licenses/by/4.0/

Keywords: Cholangiocarcinoma, Prognosis, Weighted gene correlation network analysis, Progression, Cho

Funding: The authors received no funding for this work.

==============================
Background

As the most frequently occurred tumor in biliary tract, cholangiocarcinoma (CCA) is mainly characterized by its late diagnosis and poor outcome. It is therefore urgent to identify specific genes and pathways associated with its progression and prognosis.

Materials and Methods

The differentially expressed genes in The Cancer Genome Atlas were analyzed to build the co-expression network by Weighted gene co-expression network analysis (WGCNA). Gene ontology (GO) as well as Kyoto Encyclopedia of Genes and Genomes (KEGG) analysis were conducted for the selected genes. Module–clinical trait relationships were analyzed to explore the association with clinicopathological parameters. Log-rank tests and cox regression were used to identify the prognosis-related genes.

Results

The most related modules with CCA development were tan module containing 181 genes and salmon module with 148 genes. GO analysis suggested enrichment terms of digestion, hormone transport and secretion, epithelial cell proliferation, signal release, fibroblast activation, response to acid chemical, wnt, Nicotinamide adenine dinucleotide phosphate metabolism. KEGG analysis demonstrated 15 significantly altered pathways including glutathione metabolism, wnt, central carbon metabolism, mTOR, pancreatic secretion, protein digestion, axon guidance, retinol metabolism, insulin secretion, salivary secretion, fat digestion. Key genes of SOX2, KIT, PRSS56, WNT9A, SLC4A4, PRRG4, PANX2, PIR, RASSF8, MFSD4A, INS, RNF39, IL1R2, CST1, and PPP3CA might be potential prognostic markers for CCA, of which RNF39 and PRSS56 also showed significant correlation with clinical stage.

Discussion

Differentially expressed genes and key modules contributing to CCA development were identified by WGCNA. Our results offer novel insights into the characteristics in the etiology, prognosis, and treatment of CCA.

Introduction

Cholangiocarcinoma (CCA), characterized by cholangiocyte differentiation, represents a type of aggressive tumor which derive form lesions in different parts of the biliary tree (Bergquist & Von Seth, 2015). As the most frequently occurred biliary tumor and the second after liver cancer, CCA is mainly characterized by its late diagnosis and poor outcome (Blechacz, 2017). Currently, it is estimated that overall 5-year survival rate of CCA is less than 10% and only approximately one third of the CCA patients are suitable for curative treatment at the time of diagnosis (Doherty, Nambudiri & Palmer, 2017). Therefore, it is urgently required to identify specific genes and pathways associated with its initiation, progression, and prognosis.

The common category of CCA are on basis of different locations of intrahepatic, perihilar, or distal CCA (Oliveira et al., 2017). Most patients are diagnosed with non-resectable status and suffer an extremely poor prognosis (Rizvi & Gores, 2017). A number of diseases have been reported to be related with the development of CCA including viral hepatitis B and C, choledocholithiasis, obesity, exposure to toxic agents, liver cirrhosis, diabetes mellitus, congenital hepatic fibrosis, and primary sclerosing cholangitis (Erichsen et al., 2009). Every year, a larger number of cases occur in Southeast Asia, in which hepatobiliary flukes Opisthorchis viverrini infection or Clonorchis sinensis infection were considered to be implicated in the development of CCA (Razumilava & Gores, 2014). Interleukin-6, a pro-inflammatory cytokine associated with downstream activation of oncogenic pathways, has been linked with CCA development (Park et al., 1999). Frequent mutations of oncogenes such as KRAS, as well as cancer suppressor genes of TP53 and SMAD4 were identified by next generation sequencing in CCA (Chan-on et al., 2013). In addition, research from several case-controlled studies has demonstrated multiple genetic polymorphisms that might be implicated in CCA carcinogenesis (Bridgewater et al., 2014).

Although a number of genes and mechanisms have been proved to be closely implicated in the development of CCA, the comprehensive picture of the whole genes and regulations of CCA is still unclear. In recent years, bioinformatic methods become increasingly effective in exploration and analysis of multiple genes or proteins of complicated diseases. Weighted gene co-expression network analysis (WGCNA), a new gene co-expression analysis method, has been successfully used to screen biomarkers and pathways that could be applied in susceptibility genes, diagnose and treatment of cancer. In this study, WGCNA was conducted to analyze data of The Cancer Genome Atlas (TCGA) data repository of CCA to screen modules and core genes in pathogenesis, progression, and survival of CCA.

Materials and Methods

Publically available data sets

RNA expression as well as clinical parameters of CCA patients were obtained from TCGA database (cancergenome.nih.gov). The level of gene expression was tested as Transcripts Per Kilobase of exon model per Million mapped reads. Clinical characteristics had the sample type, histology grade, recurrence, histologic grade, and prognosis. Each sample must had complete pathology stage and histology records. If the expression of genes showed limited variation, we regraded them as noise and discard these ones because the results of these genes might come from systematic error and have limited significance.

Construction of co-expression network of genes

In this study, we adopted WGCNA method to build a co-expression network for certain genes using R language (Langfelder & Horvath, 2008). We used WGCNA method to calculate power number in order to construct modules through co-expression. WGCNA method was also performed for construction of the co-expression network and extraction of the genetic information in the most relevant module. “Heatmap tool” package of R software was selected to analyze the correlation degree among modules. As a representative of the gene expression profiles of a module, module eigengene (ME) was used to evaluate the relationship between module and overall survival.

Identification of clinical traits-related modules

We then performed Pearson’s correlation examination to explore the relation of MEs with clinical traits of histologic grade, recurrence, pathologic M, pathologic N, pathologic T. A P-value less than 0.05 indicated statistical significance. The two most significant modules are selected as hub modules.

GO and KEGG analysis

In the present study, we adopted the clusterprofiler package of R language to investigate the possible biological procedures and signaling pathways of selected genes in different module. Gene ontology (GO) analysis (Ashburner et al., 2000) as well as Kyoto Encyclopedia of Genes and Genomes (KEGG) analysis were conducted (Kanehisa et al., 2004), which enable computational prediction of higher-level complexity of cellular processes and organism behavior from genomic information.

Prognostic value of genes of hub modules

Prognosis investigation was conducted using the “survival” package within R software. Cox regression test was used to calculate the hazard ratio and its 95% confidence interval while Kaplan–Meier method was adopted to draw survival curve. The increased and decreased expression level of each gene in the candidate module was decided according to median value.

Screening for candidate hub genes

We chose genes associated with prognosis as candidate genes. Next, we analyzed the association between candidate genes and stage parameter. The genes associated with both overall survival and clinical stage were selected as hub genes.

Results

Gene expression profile of CCA

Clinical characteristics and RNA expression value for 33 CCA individuals were downloaded at TCGA. Altogether 6,417 relatively more variant genes on the basis of median absolute deviation value were considered in our subsequent investigaiton. When soft thresholding power β came to 7, the connectivity amone genes accord with a scale-free network distribution (Fig. S1). A total of 21 modules were screened using hierarchical clustering along with Dynamic branch Cutting by WGCNA analysis. And each different color represent a sole module. Interaction relationship analysis of co-expression genes was shown in Fig. 1. The gene counts within modules were from 37 to 909. In this study, the gray module collected a series of genes which did not belong to any module.

Figure 1 Interaction relationship analysis of co-expression genes in cholangiocarcinoma.

Association of module with clinical traits

Identification of genes and modules which associate with clinical characteristic would greatly help the understanding the underlying mechanisms of certain traits. In our analysis, the clinical parameters of CCA of histologic grade, recurrence, pathologic M, pathologic N, pathologic T were selected in the module–trait relationship analysis. As was suggested in Fig. 2, the two relatively most significant modules of tan and salmon were selected as hub modules because their significant relation with clinical parameters including grade and T, N, M of CCA.

Figure 2 The module–clinical trait relationships of genes involved in clinicopathological parameters of cholangiocarcinoma patients.

Enrichment investigation of the hub modules

We next proceeded GO and KEGG investigation of the genes within tan module and salmon module. Altogether 50 terms showed significant difference in GO enrichment analysis (Table S1) and the most important terms were summarized in Table 1. As was illustrated in Fig. 3, the salmon module in GO analysis was related with digestion, hormone transport, hormone secretion, epithelial cell proliferation, regulation of hormone levels, signal release while the tan module was associated with fibroblast activation, response to acid chemical, wnt signaling pathway, cell-cell signaling by wnt, Nicotinamide adenine dinucleotide phosphate (NADP) metabolic process, negative regulation of neuron differentiation, negative regulation of cell development. According to the KEGG analysis, 15 pathways were significantly altered in hub modules including glutathione metabolism, wnt signaling pathway, central carbon metabolism in cancer, mTOR signal, pancreas secretion, protein digestion or absorption, axon guidance, retinol metabolism, insulin secretion, salivary secretion, and fat digestion and absorption (Fig. 4; Table 2).

Figure 3 Gene Ontology analysis for genes in the hub modules of tan module and salmon module in cholangiocarcinoma.

Figure 4 The correlation between the expression levels of key genes of hub modules and the survival of CAA patients.

Table 1 Gene ontology (GO) analysis of the genes involved in hub modules.

Module	ID	Description	GeneRatio	P-value	Count	
tan	GO:0003179	Heart valve morphogenesis	5/181	3.36E-05	5	
tan	GO:0006739	NADP metabolic process	5/181	3.36E-05	5	
tan	GO:0003170	Heart valve development	5/181	5.67E-05	5	
tan	GO:0003281	Ventricular septum development	6/181	5.83E-05	6	
tan	GO:0060411	Cardiac septum morphogenesis	6/181	6.92E-05	6	
tan	GO:0003279	Cardiac septum development	7/181	7.37E-05	7	
tan	GO:0072537	Fibroblast activation	3/181	0.00013	3	
tan	GO:0001101	Response to acid chemical	12/181	0.00015	12	
tan	GO:0016055	Wnt signaling pathway	15/181	0.00016	15	
tan	GO:0198738	Cell-cell signaling by wnt	15/181	0.00017	15	
salmon	GO:0007586	Digestion	16/148	1.02E-12	16	
salmon	GO:0010817	Regulation of hormone levels	22/148	1.53E-10	22	
salmon	GO:0009914	Hormone transport	15/148	7.10E-08	15	
salmon	GO:0046879	Hormone secretion	14/148	2.85E-07	14	
salmon	GO:0023061	Signal release	16/148	6.22E-07	16	
salmon	GO:0030072	Peptide hormone secretion	12/148	1.16E-06	12	
salmon	GO:0046883	Regulation of hormone secretion	12/148	1.84E-06	12	
salmon	GO:0050679	Positive regulation of epithelial cell proliferation	10/148	2.07E-06	10	
salmon	GO:0090276	Regulation of peptide hormone secretion	10/148	8.73E-06	10	
salmon	GO:0050673	Epithelial cell proliferation	13/148	1.32E-05	13	

Table 2 Kyoto Encyclopedia of Genes and Genomes (KEGG) pathways analysis of the genes involved in hub modules.

Module	ID	Description	GeneRatio	P-value	Count	
tan	hsa05224	Breast cancer	10/81	3.65E-06	10	
tan	hsa00480	Glutathione metabolism	6/81	2.84E-05	6	
tan	hsa04310	Wnt signaling pathway	8/81	0.00017	8	
tan	hsa05230	Central carbon metabolism in cancer	5/81	0.00066	5	
tan	hsa05226	Gastric cancer	7/81	0.00111	7	
tan	hsa04150	mTOR signaling pathway	7/81	0.0012	7	
salmon	hsa04972	Pancreatic secretion	11/64	3.74E-10	11	
salmon	hsa04974	Protein digestion and absorption	10/64	3.45E-09	10	
salmon	hsa05217	Basal cell carcinoma	3/64	0.01654	3	
salmon	hsa04360	Axon guidance	5/64	0.0167	5	
salmon	hsa00830	Retinol metabolism	3/64	0.01948	3	
salmon	hsa04380	Osteoclast differentiation	4/64	0.02372	4	
salmon	hsa04911	Insulin secretion	3/64	0.03605	3	
salmon	hsa04970	Salivary secretion	3/64	0.04162	3	
salmon	hsa04975	Fat digestion and absorption	2/64	0.04786	2	

Prognosis analysis

Since the tan and salmon module were selected from as the hub modules, we are interested in whether the genes with in the modules might predict the survival of CCA patients. Of the 358 genes, a total of 15 genes (SOX2, KIT, PRSS56, WNT9A, SLC4A4, PRRG4, PANX2, PIR, RASSF8, MFSD4A, INS, RNF39, IL1R2, CST1, and PPP3CA) demonstrated significant association with the prognosis of CCA (Table 3). The survival curve for the top four genes in salmon module (INS: HR = 4.670, P = 0.006; RNF39: HR = 2.811, P = 0.032; CST1: HR = 2.846, P = 0.017; PPP3CA: HR = 3.151, P = 0.010) and the top four gene in tan module (PRSS56: HR = 2.876, P = 0.018); SLC4A4: HR = 0.352, P = 0.025; PRRG4: HR = 0.339, P = 0.023; PIR: HR = 2.701, P = 0.027) were illustrated in Fig. 5. In addition, key genes of DNA repair and immune regulation were analyzed in relation to CCA prognosis, the results of which suggested that DNA repair gene XPC (HR = 0.369, P = 0.048) and immune regulator CD28 (HR = 0.364, P = 0.045) were related with favorable survival of CCA (Table S2). We also analyzed the prognosis of genes more frequently implicated in CCA including KRAS, TP53, BAP1. The mutation frequency of KRAS, TP53, BAP1 were 5.71%(2), 8.57%(3), 5.71%(2) out of 33 CCA patients. And no significant association with prognosis was observed.

Figure 5 The correlation between the expression levels of key genes of hub modules and the survival of CAA patients.

(A) INS; (B) PPP3CA; (C) CST1; (D) RNF39; (E) PRSS56; (F) PRRG4; (G) SLC4A4; (H) PIR.

Table 3 Key genes associated with prognosis of cholangiocarcinoma patients in hub modules.

Gene	Full name	Module	HR	95% CI	P-value	
SOX2	SRY-box 2	tan	0.382	[0.145–1.002]	0.043	
KIT	KIT proto-oncogene receptor tyrosine kinase	tan	0.341	[0.132–0.883]	0.048	
PRSS56	Serine protease 56	tan	2.876	[1.048–7.892]	0.018	
WNT9A	Wnt family member 9A	tan	0.389	[0.144–1.050]	0.041	
SLC4A4	Solute carrier family 4 member 4	tan	0.352	[0.133–0.931]	0.025	
PRRG4	Proline rich and Gla domain 4	tan	0.339	[0.127–0.900]	0.023	
PANX2	Pannexin 2	tan	2.600	[0.963–7.022]	0.037	
PIR	Pirin	tan	2.701	[0.994–7.337]	0.027	
RASSF8	Ras association domain family member 8	tan	0.396	[0.147–1.065]	0.040	
MFSD4A	Major facilitator superfamily domain containing 4A	salmon	0.371	[0.143–0.962]	0.043	
INS	Insulin	salmon	4.670	[0.458–47.564]	0.006	
RNF39	Ring finger protein 39	salmon	2.811	[1.063–7.434]	0.032	
IL1R2	Interleukin 1 receptor type 2	salmon	0.379	[0.144–0.997]	0.041	
CST1	Cystatin SN	salmon	2.846	[1.039–7.797]	0.017	
PPP3CA	Protein phosphatase 3 catalytic subunit alpha	salmon	3.151	[1.129–8.793]	0.010	

Identification of hub genes associated with both stage and survival

We then analyze the association between clinical stage and gene which showed significant relation with prognosis. As was shown in Fig. 6, two genes (RNF39 and PRSS56) showed difference between two groups (Table 4).

Figure 6 The differential expression of potential hub genes in different stages of CCA patients.

(A) Differential expression of INS, PPP3CA, CST1 and RNF39; (B) Differential expression of PRSS56, PRRG4, SLC4A4 and PIR.

Table 4 The association of prognosis-related genes with clinical of stage of cholangiocarcinoma patients.

Gene symbol	Gene title	Position	OR	P	
MFSD4A	Major facilitator superfamily domain containing 4A	1q32.1	1.036 (0.988–1.093)	0.155	
SOX2	SRY-Box 2	3q26.33	1.037 (0.986–1.115)	0.184	
INS	Insulin	11p15.5	1.125 (0.996–1.320)	0.059	
KIT	KIT proto-oncogene receptor tyrosine kinase	4q12	1.033 (0.991–1.091)	0.146	
PRSS56	Serine protease 56	2q37.1	1.162 (1.002–1.431)	0.013	
RNF39	Ring finger protein 39	6p22.1	1.069 (1.006–1.167)	0.019	
WNT9A	Wnt family member 9A	1q42.13	0.965 (0.904–1.018)	0.235	
SLC4A4	Solute carrier family 4 member 4	4q13.3	1.001 (0.973–1.029)	0.945	
IL1R2	Interleukin 1 receptor type 2	2q11.2	1.027 (0.993–1.068)	0.131	
CST1	Cystatin SN	20p11.21	1.007 (0.988–1.026)	0.490	
PRRG4	Proline rich and Gla domain 4	11p13	0.983 (0.922–1.041)	0.574	
PANX2	Pannexin 2	22q13.33	0.987 (0.940–1.024)	0.517	
PIR	Pirin	Xp22.2	0.983 (0.930–1.027)	0.489	
RASSF8	Serine protease 8	12p12.1	0.942 (0.863–1.002)	0.094	
PPP3CA	Protein phosphatase 3 catalytic subunit alpha	4q24	1.021 (0.960–1.086)	0.492	
Note:

The bold values indicate significant association between gene expression and clinical stage of CCA patients.

Discussion

Although surgical therapy and liver transplantation might be possible options for certain patients who suffer CCA, the 5-year survival ratio of CCA remain extremely low. Comprehensive exploration of the genetic and epigenetic profiles as well as their complicated interactions with environment would greatly help the treatment and survival for CCA patients. Until now, our understanding of the complex mechanisms of CCA is still limited. One study using CCA data also analyzed the genes implicated in CCA (Tian et al., 2019). They dissected the genome network by other bioinformatic tool to explore the protein–protein interaction. In this study, we provided our own gene lists with significance by WGCNA. Besides, we adopted KEGG analysis to explore the biological functions for the tumor progression. A differentially expression analysis followed by WGCNA to identify genes and pathways related were performed to the clinical parameters and prognosis of CCA. In addition, enrichment analysis of the genes in core modules suggested significant involvement of pathways such as glutathione metabolism, wnt signaling, mTOR signaling, pancreatic secretion, protein digestion and absorption, insulin secretion, and salivary secretion.

In this present study, RNA sequencing data for CCA samples from TCGA were systematically analyzed. According to the WGCNA analysis of most variant genes, altogether 21 modules were identified and the two most significant modules of tan and salmon were selected as hub modules. GO analysis of salmon module demonstrated enrichment in digestion, hormone transport, hormone secretion, epithelial cell proliferation, regulation of hormone levels, signal release. It can be concluded from our analysis that abnormal hormone regulation might be closely implicated in CCA progression. Previously, autocrine parathyroid hormone-like hormone was indicated to promote intrahepatic CCA cell growth through enhanced ERK/JNK-ATF2-cyclinD1 signaling (Tang et al., 2017). There were also reports of CCA patients with aberrant levels of parathyroid hormone-related protein (Matsumoto et al., 2014; Ozawa et al., 2017). From this point of view, future investigations concerning the synthesis, transport and secretion of relevant hormones might provide novel insight into CCA development.

In addition, the tan module was associated with fibroblast activation, response to acid chemical, wnt signaling pathway, NADP metabolic process, negative modulation of neuron differentiation, negative modulation of cell development. Wnt signaling pathway has been proved to be involved in the growth and progression of various types of cancer (Clevers & Nusse, 2012). Previously, enhanced wnt signaling has been found to be a characteristic of CCA (Boulter et al., 2015) which related with metastasis or chemoresistance (Wang et al., 2015). Besides, lncRNA PCAT1 regulated the development of extrahepatic CCA progression by Wnt and β-catenin signaling pathway (Zhang et al., 2017b). It is worth noting that response to acid chemical was also a hallmark in CCA development according to our study. Indeed, glycocholic acid and taurochenodeoxycholic acid were found to be potential phenotypic biomarkers in CCA (Song et al., 2018). Conjugated bile acids were suggested to promote invasive growth of CCA cells via activation of sphingosine 1-phosphate receptor 2 (Liu et al., 2014). The role of other biological processes such as NADP metabolic process and neuron differentiation required further studies to elucidate.

According to the KEGG analysis, 15 pathways were significantly altered in hub modules including glutathione metabolism, wnt signaling pathway, central carbon metabolism in cancer, mTOR signaling pathway, pancreatic secretion, protein digestion and absorption, axon guidance, retinol metabolism, insulin secretion, salivary secretion, and fat digestion and absorption. Some of the identified pathways have been suggested to play critical role in CCA progression and therapy. In intrahepatic cholangiocarcinoma (ICC) mouse model, investigators proved that mTOR kinase inhibitors inducing strong ICC cell apoptosis and may be beneficial for the treatment of ICC (Zhang et al., 2017a). It has been reported that the contact inhibition of CCA cells could be overcome by c‑Myc via the mTOR pathway (Luo et al., 2017). The identified significant pathways of secretion and absorbance of multiple digestive juice remind us the importance of digestive balance in the development of CCA, which required further investigations to follow. The significant altered pathways of insulin secretion, salivary secretion, and fat digestion and absorption confirmed the key role of digestive regulation in CCA, as was indicated by a number of previous investigations (Chaiyarit et al., 2011; Chen et al., 2013; Rizvi et al., 2014).

The 5-year survival rate of CCA currently remains unsatisfactory despite recent improvements in treatments of surgical resection and chemotherapy (Squadroni et al., 2017). The selected genes hold great possibilities to become potential biomarkers for the prediction of survival of CCA patients. We eventually screened 15 prognosis-related genes in CAA after analyzing the hub modules of tan and salmon. Increased expression of SOX2 in CCA cells lead to cell proliferation, decreased cell apoptosis, and increased cell invasion and metastasis (Sun et al., 2014). Besides, SOX2 expression correlated with aggressive behavior and worse overall survival in ICC (Gu & Jang, 2014). Among the identified prognosis-related genes, RNF39 and PRSS56 also showed significant correlation with clinical stage. Until now, little is known about the functions of these two genes in cancer. RNF39 encoded a member of ring finger proteins, which was studied in Behcets disease, a chronic inflammatory autoimmune disease (Kurata et al., 2010). In addition, RNF39 protein was suggested to participate in the replication process according to both studies on HIV-1 individuals and cell-lines (Lin et al., 2014). PRSS56 has been established as an effective marker of adult neurogenesis in the brain of mouse (Jourdon et al., 2016). Variants in PRSS56 may lead to primary angle-closure glaucoma and high hyperopia (Jiang et al., 2013). Whether RNF39 and PRSS56 could be a robust biomarker for CCA and their underlying mechanisms still required further studies to explore. In addition, DNA repair gene XPC and immune regulator CD28 were related with favorable survival of CCA, which indicate probable implication of DNA repair and immune response in CCA development. Because no distant metastasis information was available in TCGA data of CCA, we did not perform association analysis of genes with CCA metastasis, which require future studies to clarify.

Conclusion

In summary, differentially expressed genes and key modules contributing to progression of CCA were identified by means of WGCNA. Pathways including glutathione metabolism, wnt signaling, central carbon metabolism in cancer, mTOR signaling, pancreatic secretion, protein and fat digestion, insulin secretion, and salivary secretion might be closely related with CCA development. Key genes such as RNF39 and PRSS56 might be potential prognostic markers for CCA. The results offer novel insights into the understanding of the development and prognosis of CAA.

Supplemental Information

Supplemental Information 1 Threshold selection of WGCNA analysis.

Click here for additional data file.

Supplemental Information 2 All 50 terms which showed significant difference in GO enrichment analysis.

Click here for additional data file.

Supplemental Information 3 Prognostic value of key genes of DNA repair and immune regulation.

Click here for additional data file.

Additional Information and Declarations

Competing Interests

Author Contributions

Data Availability

The authors declare that they have no competing interests.

Jingwei Liu performed the experiments, analyzed the data, contributed reagents/materials/analysis tools, prepared figures and/or tables, authored or reviewed drafts of the paper, approved the final draft.

Weixin Liu conceived and designed the experiments, authored or reviewed drafts of the paper, approved the final draft.

Hao Li performed the experiments, analyzed the data, contributed reagents/materials/analysis tools, authored or reviewed drafts of the paper, approved the final draft.

Qiuping Deng performed the experiments, contributed reagents/materials/analysis tools, prepared figures and/or tables, approved the final draft.

Meiqi Yang performed the experiments, contributed reagents/materials/analysis tools, prepared figures and/or tables, approved the final draft.

Xuemei Li performed the experiments, contributed reagents/materials/analysis tools, prepared figures and/or tables, approved the final draft.

Zeng Liang performed the experiments, contributed reagents/materials/analysis tools, prepared figures and/or tables, approved the final draft.

The following information was supplied regarding data availability:

Raw data is available at TCGA (cancergenome.nih.gov), search term “TCGA-CHOL.”

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
