# Peer review of "Identification of key genes and pathways associated with cholangiocarcinoma development based on weighted gene correlation network analysis"

_PeerJ, doi:10.7717/peerj.7968_

## Round 0.1 · original submission · Minor Revisions

Overall I like the way the authors have presented the paper with a fairly reasonable approach to their methods. I do recommend the authors to expand on their methods.

1. Why did the authors use salmon and tan modules in GO analysis?
2. May consider reporting the frequency altered gene expression for major genes as BAP1, TP53, KRAS
3. Expand on abbreviations prior to using them

I do suggest the authors to expand upon the introduction for this article with a clinical implications for this study and the studies in the past that have highlighted similar test results.

Good effort, certainly worth publishing. All the best.

Reviewer 1 ·

Basic reporting

The language is in general good. The rules for the tables, datas, formatting are proper followed. However, the authors can improve the clarity of the paper.

The author wrote very general introduction about CCA. It will be relevant to review the current progress of the genetics and bioinformatic analysis for CCA.

The same with the discussion part. The novelty of WNT signaling is limited so please write it simple. It will be good to write on the secretion signaling and also discuss function of the genes.

I would like to see the authors illustrate the reason to use tan and salmon module in GO analysis.

Please write full name of any abbreviation when mentioned first time. Please provide the full name and abbreviation of WGCNA in line 28. Same thing with “TCGA”.

Experimental design

The authors attempted to identify potential biomarkers highly associated with tumor progression. However, the results may only be biomarkers for late stages but not the early detectors for CCA due to the author’s design. Since they used the information after the tumor initiated, comparing Stage I-II with Stage III-IV, their result can be metastasis makers or late progression markers. The later only has limited clinical significance.

Biomarkers for metastasis is of great significance clinically. Nevertheless, the TNM system combined information of the metastasis and tumor orthotopic growth. The TNM system might not be the best trait to identify such markers. I will like to see the analysis to specifically identify genes with a more specific biological function, such as metastasis. This is better for follow-up studies too to link the genes with biological function.

The TCGA database need to be proper sited. There are at least two TCGA datasets. The author needs to specify the set.

Furthermore, the authors analysis 33 patients but there are additional samples in PanCancer Atlas or provisional sets. The author needs to list the reason for such selection or include all the samples.

Validity of the findings

Recently, a paper with the similar approach was published (Mao et al. Hepatology Res. June 2019). The originality is thus limited here. However, Mao et al. dissect the genome network by other bioinformatic tool to explore the protein-protein interaction. On the other hand, the authors adopted Kyoto Encyclopedia of Genes and Genomes (KEGG) analysis to explore the biological functions for the tumor progression. The result putative gene list is very different within two papers. Therefore, Liu et al. still brought valuable information for cancer-driven genes of CCA.

Besides, in the draft the authors need to mentioned the above paper and compare it with the author's study.

Additional comments

Weighted correlation network analysis (WGCNA) is an established gene co-expression analysis method. Liu et al. applied WGCNA to the data from The Cancer Genome Atlas (TCGA). They identified several genes as putative biomarkers for cholangiocarcinoma (CCA).

Reviewer 2 ·

Basic reporting

No issues

Experimental design

-"If the expression of genes showed limited variation, we regraded them as noise and discard these ones."; I'm unclear as to why this was done
-"We chose genes associated with prognosis as candidate genes"; Cant say I agree with this. I'd rather see it more open ended regarding the selection of genes. I'd rather see the prognostic of genes more frequently implicated in CCA, not necessarily genes we may / may not encounter
-I'm not sure how the tan and salmon modules were specifically made. This part is unclear in both the methods and results. The Gene Ontology and KEGG methods need to be better defined and described

Validity of the findings

-The concept of modules based on GO does not make much sense, given the epigenetic factors which may be at play and the lack of commonality between some of the genes in the same modules. What would be better would be to report on the individual genes themselves. Additionally focusing on gene expression beyond the GO clusterings would be far more implicating. For example reporting on the frequency altered gene expression for major genes as BAP1, TP53, KRAS would be far more interesting and practical.
-If modules were to be used what would be needed would be the exact frequency of these mutations detailed in TCGA data. It is not clear how frequently mutations (ie SOX2, KIT) and altered gene expression (of these proteins) are encountered in CCA to begin with.
-I would also be interested in knowing the prognostic and expression profiles other other modules such as DNA repair and immune based mechanisms. Given the therapeutic implications involved in these, these need to be described

Additional comments

-GO and KEGG analysis need to be better described. It is unclear based on the study exactly how the submodules were mapped
-Focus should be on protein expression of major genes which are more frequently implicated in CCA (ie TP53, BAP1). Unclear of the frequency of some of these mutations and through what means their expression is altered (ie epigenetic regulators)

Bottom Line: Overall interesting article. I'm not totally convinced of the findings, given the unclear frequency of some of these alterations and possible other factors contributing to altered gene expression in some of these cases. What I would be more interested in would be prognostic implications of genes tied to major mutations in CCA (ie KRAS, TP53, BAP1 etc) and gene expression signatures involving immune regulatory genes and DNA damage regulating genes, given potential therapeutic implications for both. I'm sure the data is there it just needs to be tailored to make an interesting case for prognosis and potential treatments for CCA.

---

## Round 0.2 · accepted · Accept

Acceptable edits, all the best.

Reviewer 1 ·

Basic reporting

The writing is clear with professional English.


My comments for the intro and discussion section have been covered.

Experimental design

The author has explained the current knowledge gap. In the current version, the authors provide details about the methods, specifically about the GO and KEGG.

Validity of the findings

The authors have added more information about the current progress for the analysis of TCGA data of CCA. It is sufficient for publication

Additional comments

The authors have addressed all my comments. The paper meets the criteria of PeeJ Review